# *p*-Coumaric Acid Enhances Hypothalamic Leptin Signaling and Glucose Homeostasis in Mice via Differential Effects on AMPK Activation

**DOI:** 10.3390/ijms22031431

**Published:** 2021-01-31

**Authors:** Linh V. Nguyen, Khoa D. A. Nguyen, Chi-Thanh Ma, Quoc-Thai Nguyen, Huong T. H. Nguyen, Dong-Joo Yang, Trung Le Tran, Ki Woo Kim, Khanh V. Doan

**Affiliations:** 1School of Medicine, Tan Tao University, Duc Hoa 850000, Long An, Vietnam; linh.nguyen@ttu.edu.vn (L.V.N.); khoa.nguyen@ttu.edu.vn (K.D.A.N.); 2Faculty of Pharmacy, University of Medicine and Pharmacy, Ho Chi Minh 700000, Vietnam; mcthanh@ump.edu.vn (C.-T.M.); nqthai@ump.edu.vn (Q.-T.N.); 3School of Pharmacy, Van Lang University, Ho Chi Minh 700000, Vietnam; huong.nth@vlu.edu.vn; 4Departments of Oral Biology and Applied Biological Science, BK21 Plus, Yonsei University College of Dentistry, Seoul 03722, Korea; YDJ1991@yuhs.ac (D.-J.Y.); TRUNGTRANLE@yuhs.ac (T.L.T.)

**Keywords:** *p*-Coumaric acid, AMPK, leptin signaling, glucose homeostasis, HFD-induced obesity

## Abstract

AMP-activated protein kinase (AMPK) plays a crucial role in the regulation of energy homeostasis in both peripheral metabolic organs and the central nervous system. Recent studies indicated that *p*-Coumaric acid (CA), a hydroxycinnamic phenolic acid, potentially activated the peripheral AMPK pathway to exert beneficial effects on glucose metabolism in vitro. However, CA’s actions on central AMPK activity and whole-body glucose homeostasis have not yet been investigated. Here, we reported that CA exhibited different effects on peripheral and central AMPK activation both in vitro and in vivo. Specifically, while CA treatment promoted hepatic AMPK activation, it showed an inhibitory effect on hypothalamic AMPK activity possibly by activating the S6 kinase. Furthermore, CA treatment enhanced hypothalamic leptin sensitivity, resulting in increased proopiomelanocortin (POMC) expression, decreased agouti-related peptide (AgRP) expression, and reduced daily food intake. Overall, CA treatment improved blood glucose control, glucose tolerance, and insulin sensitivity. Together, these results suggested that CA treatment enhanced hypothalamic leptin signaling and whole-body glucose homeostasis, possibly via its differential effects on AMPK activation.

## 1. Introduction

AMP-activated protein kinase (AMPK) is a cellular energy sensor that plays a crucial role in the regulation of whole-body energy balance [1]. Activation of AMPK is mediated via several mechanisms but requires phosphorylation of Thr172 residue in the catalytic α-subunit [1]. It is well known that AMPK activation in peripheral metabolic organs such as liver, muscle, and adipose tissues shifts the cellular metabolism from anabolic to catabolic processes, resulting in many favorable effects on glucose and lipid metabolism and insulin sensitivity [1,2]. However, in the central nervous system, activation of hypothalamic AMPK was shown to promote feeding and weight gain whereas inhibition of hypothalamic AMPK activity enhanced hypophagia and weight loss, as well as lowering glucose production [3,4,5]. Modulation of AMPK activity in the hypothalamus was also shown to mediate leptin’s effects on food intake and whole-body energy metabolism [3,6]. The pleiotropic actions of AMPK in metabolic health make it a promising therapeutic target in the treatment of obesity and metabolic diseases [7,8,9].

Besides physiological upstream kinases and adenosine nucleotides (AMP, ADP, ATP), AMPK activity can be modulated by pharmacological agents such as the clinically therapeutic agent metformin and several natural plant products [10]. Although the structures of these natural compounds are varied, most of them can be classified as polyphenols, which are present ubiquitously in plants and possess various bioactivities such as antioxidative, anti-inflammatory, antibacterial, antidiabetic, and antiproliferative effects [10,11]. Among the natural polyphenols, resveratrol and several flavonoids have been extensively investigated for the ability to activate the AMPK pathway and their beneficial metabolic effects [12]. However, mechanistic insights into AMPK activation and metabolic actions of phenolic acids that are other non-flavonoid polyphenols have only recently been explored.

*p*-Coumaric acid (4-hydroxycinnamic acid, CA) is a natural phenolic acid found in many kinds of plants—either in free or conjugated form—and displays various bioactivities, such as antioxidant, anti-inflammatory, and anticancer properties [13]. CA treatment has also been reported to have protective effects against hyperlipidemia and streptozocin-induced diabetes [14,15]. Recent evidence suggests that CA might activate the AMPK pathway in several peripheral cells/tissues to exert beneficial metabolic effects [16,17,18]. However, mechanistic insights into CA effects on central AMPK and leptin sensitivity have not yet been investigated. In the present study, we employed the cell culture system and high-fat diet (HFD)-induced obese mouse model to investigate CA’s effects on AMPK activation and hypothalamic leptin signaling, as well as whole-body glucose homeostasis.

## 2. Results

### 2.1. p-Coumaric Acid Exhibited Differential Effects on Peripheral and Central AMPK Activation

*p*-Coumaric acid (CA) or plant extracts containing CA have been shown to activate the AMPK pathway in several metabolic cell lines [16,17,18]. Using the therapeutic agent metformin which is well known to activate the AMPK prominently in hepatocytes [19,20], we first investigated CA’s effect on AMPK activation in the hepatic HepG2 cells [21]. Consistent with previous study, we found that CA treatment potently activated AMPK in a dose-dependent manner in HepG2 cells at an optimal concentration of 20 μM (Figure 1A,B and Appendix A) [18]. Because metformin has been shown to display differential effect on AMPK activation in the hypothalamic neurons to exert its anorexigenic actions [22,23], we next used neuronal hypothalamic N1 cell culture to explore CA effect on central AMPK activation [24,25]. Interestingly, we found that CA treatment significantly inhibited AMPK activation in neuronal hypothalamic N1 cells, similar to metformin’s action reported previously (Figure 1C,D and Appendix A) [22]. The inhibitory effect of CA treatment on AMPK in the neuronal hypothalamic N1 cells seemed maximal at 10 μM (Appendix A).

Next, we further investigated CA’s effects on peripheral and central AMPK activation in vivo. Mice were administered either CA or metformin at doses of 200 mg/kg body weight via oral gavage, and metabolic organs including livers and hypothalami were examined for AMPK activation. Based on the pharmacokinetic data of CA in rodents reported in previous studies, this dose of oral treatment would produce steady-state plasma CA concentrations comparable with those used in cell culture experiments [26,27,28]. Consistent with the results observed in cell lines, we found that CA treatment markedly activated AMPK in mice livers (Figure 2A,B). In contrast, CA treatment significantly inhibited AMPK activation in the hypothalamic samples (Figure 2C,D). Together, these results indicated that CA treatment activated hepatic AMPK, while it inhibited hypothalamic AMPK activity both in vitro and in vivo.

We next figured out the molecular mechanism underlying CA treatment inhibition of central AMPK activity. In hepatocytes, metformin was shown to inhibit mTOR/S6 kinase signaling via AMPK activation [29]. However, in the hypothalamus, metformin has been reported to activate central S6 kinase and inhibit AMPK activation [30]. Moreover, it has been shown that S6 kinase can form a complex with AMPK and cause phosphorylation on its ser491 residue, resulting in a reduction of AMPK activity [6,31]. We therefore postulated that CA treatment might also affect AMPK activation by modulating the S6 kinase activity. Interestingly, we found that CA treatment increased phosphorylated forms of p70 S6 kinase (p-p70 S6) in the hypothalamic neuronal N1 cells and in the hypothalamic samples of treated mice, similar to metformin action (Figure 2E,F and Appendix A). However, unlike metformin’s inhibitory effect on the p70 S6 kinase activity in hepatocytes, CA treatment insignificantly affect p-p70 S6 levels in the HepG2 cells and in the livers of treated mice (Appendix A). These results suggested that CA treatment might inhibit the hypothalamic AMPK activity by specifically enhancing central S6 kinase activity.

### 2.2. p-Coumaric Acid Treatment Enhanced Hypothalamic Leptin Signaling

Hypothalamic AMPK and S6 kinase have been demonstrated to respond to the leptin-melanocortin signaling and regulate food intake [3]. Moreover, metformin treatment enhanced hypothalamic leptin receptor signaling and restored leptin sensitivity in HFD-induced obese rodents with leptin resistance [32,33]. This led us to hypothesize that CA treatment might also affect the functional hypothalamic leptin signaling to suppress food consumption. To address this question, we employed HFD-induced obese mouse model to induce a leptin resistance condition and examined CA’s effect on functional leptin signaling in the hypothalamus. As shown in Figure 3A, plasma leptin levels of HFD-fed mice increased markedly compared to those of NC-fed mice, indicating the presence of leptin resistance. Consistent with previous findings, metformin treatment reduced this hyperleptinemic condition and increased phosphorylated form of the signal transducer and activator of transcription 3 (STAT3), a well-known downstream effector of leptin signaling, in the hypothalamus, indicating an enhancement in hypothalamic leptin signaling (Figure 3A–C) [34,35]. Noticeably, CA treatment also significantly reduced the HFD-induced hyperleptinemia and increased p-STAT3 levels in the hypothalamus (Figure 3A–C). Furthermore, CA treatment enhanced expression of the anorexigenic neuropeptide proopiomelanocortin (*Pomc*), while it inhibited expression of the orexigenic agouti-related neuropeptide (*Agrp*), which are well-known target genes of the hypothalamic leptin-STAT3 signaling (Figure 3D) [36]. Accordingly, CA-treated mice consumed less high-fat food compared to the control vehicle-treated mice (Figure 3E). These results were highly indicative that CA treatment enhanced functional leptin signaling in the hypothalamus.

### 2.3. p-Coumaric Acid Treatment Improved Whole-Body Glucose Homeostasis

The effects of CA treatment on peripheral and central AMPK activation, hypothalamic leptin signaling, and food intake led us to hypothesize that CA treatment regulates body weight and glucose homeostasis. To address this question, we employed the HFD-induced mouse model to further investigate the metabolic effects of CA treatment. HFD-fed mice were administered either CA (200 mg/kg of body weight) or vehicle treatment via oral gavage for an 8-week period. A group of HFD-fed mice received oral metformin administration (200 mg/kg of body weight) and served as a standard treatment and a cohort of NC-fed mice played as a negative control group. Unexpectedly, we found no statistically significant difference in body weight between HFD-fed mice treated with vehicle and CA with the notion that CA-treated mice had a slightly reducing trend in body weight, especially during the first 2 weeks of HFD challenge (Figure 4A). However, mice treated with CA displayed a significant reduction in fasting blood glucose (Figure 4B) and lower plasma insulin (Figure 4C). These results suggested that CA-treated mice display better blood glucose control compared to the vehicle control group. Therefore, we next performed glucose and insulin tolerance tests (GTT and ITT) to further investigate the effects of CA treatment on glucose tolerance and insulin sensitivity. As shown in Figure 4D and E, mice treated with CA showed enhanced clearance of an intraperitoneally injected glucose load from the body in the GTT test, indicating a better glucose tolerance phenotype. Moreover, these mice showed a significant increase in the endogenous glucose disappearance over time in response to an insulin injection during the ITT (Figure 4F,G). Taken together, these results highlighted that CA treatment in mice significantly improved insulin sensitivity and whole-body glucose homeostasis.

## 3. Discussion

Many natural polyphenols, most of which are flavonoids, have been reported to activate AMPK and elicit metabolic benefits in type 2 diabetes and metabolic syndromes [12,37,38]. In the present study, we focused on *p*-coumaric acid (CA), a hydroxycinnamic phenolic acid that has recently been reported to activate AMPK in several cell lines [16,17,18], to comprehensively investigate its actions on AMPK activation and whole-body glucose homeostasis. Our study identified an interesting finding, in which CA treatment displayed differential effects on peripheral and central AMPK activation. Specifically, while CA treatment promoted peripheral AMPK activation, it showed an inhibitory effect on the hypothalamic AMPK activation. These effects of CA treatment on AMPK activation seemed to be identical to the actions of metformin observed in previous studies [22,39].

Remarkably, we characterized that the inhibition of hypothalamic AMPK under CA treatment was accompanied by an enhanced S6 kinase activity, which might propose a mechanism underlying the CA’s inhibitory effect on hypothalamic AMPK activation. In addition, effect of CA on S6 kinase seemed to be centrally specific as we found that CA treatment did not affect S6 kinase activity in HepG2 cells or in the livers of treated mice. S6 kinase in the hypothalamus has been demonstrated to phosphorylate AMPK on Ser491 residue to prevent its phosphorylation on Thr172 and thus inhibit AMPK activation [6,31]. As pharmacokinetic studies of CA in rodents indicated that CA easily crosses the cell membrane after an oral administration and can enter the brain to exert effects in the central nervous system (CNS) [26,27,28,40], there is a high possibility that an amount of CA penetrated into the CNS might activate the hypothalamic S6 kinase and inhibit the central AMPK activity. Further studies confirming the presence of CA in the cerebrospinal fluid after an oral administration may warrant this possibility.

Inhibition of hypothalamic AMPK by S6 kinase was shown to mediate leptin effects on food intake and body weight [3]. Melanocortin leptin signaling is known to act on several neuronal nuclei in the hypothalamus to modulate the expression of neuropeptides that regulate feeding behavior [41]. Specifically, activation of leptin signaling promotes POMC expression meanwhile suppresses AgRP expression, resulting in reduced food intake [35]. In this regard, increased *Pomc* and decreased *Agrp* genes expression under CA treatment could be attributed to enhanced hypothalamic leptin sensitivity as CA treatment ameliorated the HFD-induced hyperleptinemia and increased leptin/STAT3 signaling in the hypothalamus. These CAs’ effects on hypothalamic leptin sensitivity were similar to those of metformin treatment, which has been shown to enhance leptin receptor expression and restore leptin sensitivity in obese rats [32,33].

Consequently, mice treated with CA showed hypophagia and improved whole-body glucose homeostasis even though there was no significant difference in mice body weight. Consistent with our findings, Han et al. recently demonstrated that CA treatment helped to maintain glucose homeostasis via promoting thermogenesis in brown fat tissue (BAT) by activating the mTOR/S6 kinase pathway [42]. Because hypothalamic leptin signaling is well known to regulate both feeding and energy expenditure [41], increased BAT thermogenesis under CA treatment may also be linked to the CA-induced enhancement of hypothalamic leptin sensitivity observed in our study. However, Han et al. showed that CA treatment also reduced HFD-induced mouse body weight [42]. The different findings on mice’s body weight under CA treatment might be due to a difference in CA treatment duration as Han et al. treated mice with CA for 14 weeks, while in our study the mice were treated with CA for only 8 weeks. Nonetheless, enhanced hypothalamic leptin sensitivity under CA treatment observed in our study might be an important contributor to CA’s effects on metabolic phenotypes in vivo.

There is a reciprocal relationship between peripheral AMPK activation and central leptin signaling, which are both known to play crucial roles in regulation of whole-body energy homeostasis [1,41]. While peripheral AMPK activation by calorie restriction or exercise or metformin helps to ameliorate central leptin resistance [32,43], central leptin action has been shown to increase the peripheral AMPK [44]. Moreover, central leptin was shown to inhibit AMPK activity in the hypothalamus to suppress food intake and regulate body weight [3]. In this regard, beneficial effects of CA on whole-body glucose homeostasis observed in our study could be attributed to its differential effects on peripheral and central AMPK activation or central leptin signaling enhancement or combined actions. Further study using mice with defective leptin signaling may help to demonstrate the contribution of central leptin signaling in mediation of CA’s metabolic actions on whole-body glucose metabolism.

Certain questions remain about the differential effects of CA treatment on AMPK activation and central leptin signaling in our current study. For example, similar to numerous polyphenols reported to be potential AMPK activators, we could not determine the molecular mechanism causing the AMPK activation by CA [10]. Whether CA treatment causes AMPK activation via the inhibition of mitochondrial ATP synthesis similarly to other polyphenols such as resveratrol, quercetin, and berberine warrants further study. Moreover, similar to the case of metformin, how CA activated AMPK in the periphery while inhibiting AMPK activation in the hypothalamus is not yet understood. Nevertheless, our study highlighted the metabolic actions of CA on central leptin signaling and whole-body glucose homeostasis, possibly via the modulation of AMPK activation.

## 4. Materials and Methods

### 4.1. Materials

We purchased Gibco^®^ Dulbecco’s modified Eagle’s medium (DMEM), Gibco^®^ fetal bovine serum (FBS), and Gibco^®^ penicillin-streptomycin (P/S) from Thermo Scientific (Thermo Fisher Scientific Inc., Waltham, MA, USA). *p*-Coumaric acid (Cat. No C9008) was obtained from Sigma (St. Louis, MO, USA). Metformin was purchased from TOKU-E (Bellingham, WA, USA). Primary antibodies including AMPKα (Cat. No. 2532, dilution 1:5000), phosphor-AMPKα Thr172 (Cat. No. 2531, dilution 1:5000), p70 S6 (Cat. No. 9202, dilution 1:5000), phosphor-p70 S6 Thr389 (Cat. No. 9204, dilution 1:5000), Stat3 (Cat#.No. 9139, 1:5000) and phosphor-Stat3 (Tyr705) (Cat#.No. 9131, 1:5000) were purchased from Cell Signaling (Cell Signaling Technology Inc., Danvers, MA, USA). GAPDH antibody (Cat. No. GTX100118, dilution 1:10,000) was purchased from GeneTex (GeneTex Inc., Irvine, CA, USA). The anti-rabbit secondary antibody conjugated with horseradish peroxidase (HRP) was purchased from Thermo Scientific (Thermo Fisher Scientific Inc., Waltham, MA, USA). Protease and phosphatase inhibitor cocktail tablets were purchased from Thermo Fisher Scientific (Thermo Fisher Scientific Inc., Waltham, MA, USA) and Roche (F. Hoffmann-La Roche Ltd., Basel, Switzerland), respectively. All other reagents were purchased from Intron Biotechnology (Intron Biotechnology Co., Ltd., Gyeonggi-do, Korea) unless otherwise stated. Stock and working solutions of metformin were prepared in water. Stock solutions of *p*-Coumaric acid were prepared in dimethylsulfoxide (DMSO). *p*-Coumaric acid working solutions were prepared by diluting the stock solutions in DMSO for cell culture experiments or in phosphate buffer saline (PBS) (1/20, *v*/*v*) for oral administration in animal experiments.

### 4.2. Cell Culture and Chemical Treatment

Hepatic HepG2 and hypothalamic N1 cells were cultured in DMEM media supplemented with 10% FBS and 1% P/S in a humidified atmosphere of 5% CO_2_ at 37 °C. Cells were treated with metformin dissolved in water at final concentrations of 2 mM or *p*-coumaric acid dissolved in DMSO at final concentrations up to 20 μM. Cells were then harvested and lysed to determine mRNA and protein levels by real-time qPCR or Western blot techniques.

### 4.3. Animals

All animal experiments were reviewed and approved by the Institutional Animal Care and Use Committee (IACUC) under The Scientific and Academic Council of the Tan Tao University School of Medicine (Ethical permission code: 43/HDKH.TTU.2019. Date: 01/06/2019). Swiss albino male mice (4 weeks old) were purchased from Pasteur Institute (Ho Chi Minh city, Vietnam). We kept mice in temperature in the range 22–25 °C with light automatically on/off at 06:00 a.m./p.m. Mice were provided a chow diet (AniFood, Pasteur Institute-VN, 3.84 kcal/kg with 6–8% kcal from fat, NC) or a high-fat diet (Research Diets D12492, USA, 5.24 kcal/kg with 60% kcal from fat, HFD) and filtered water ad libitum. After one week of acclimation, mice were assigned to four experimental groups: NC, HFD-Veh, HFD-Met, and HFD-CA (6 mice per group). The NC group was fed a normal chow diet and received filtered water. The HFD-CA group was fed HFD and received *p*-coumaric acid (200 mg/kg) daily via oral gavage for 8 weeks. The HFD-Veh group was orally administered with the same volume of DMSO/PBS mixture solution for the same period and served as the vehicle control group. The HFD-Met group was fed HFD and received metformin (200 mg/kg) dissolved in water as reference treatment.

Fed and fasted blood glucose levels were measured as indicated. Glucose and insulin tolerance tests (GTT and ITT) were performed after 6 weeks of treatment. After the experimental period, mice were sacrificed, and the organs were collected and stored at −80 °C for further analysis.

### 4.4. Dissecting and Harvesting Hypothalamus Tissues

We sacrificed mice in normal feeding condition by decapitation after anesthetizing with ketamine (100 mg/kg body weight). After removing the cranial skull, the intact brain was collected and washed once with ice-cold 1X phosphate buffer saline (PBS). Under low-power dissecting microscope, the hypothalamus region was viewed and gently cut from the whole brain using small-sized scissors. All of the hypothalamus samples were immediately snap-frozen on dry ice and stored at −80 °C.

### 4.5. Food Intake Measurement

Experimental mice were housed individually, and the food intake was recorded daily at 8:00 and 17:00 during a 3-day period and was normalized to the initial body weight.

### 4.6. Insulin and Leptin Measurement

Plasma insulin and leptin levels were measured from the blood samples collected from tail nicks using ELISA kits (Morinaga Institute of Biological Science, Yokohama, Japan) follow the manufacturer’s instructions.

### 4.7. GTT and ITT Experiments

For GTT, mice were removed from food 18 h before experiment, with water provided ad libitum. A solution of D-glucose (1.2 g/kg body weight) in normal saline was intraperitoneally injected. Blood glucose levels at t_0_, 15, 30, 60, 90, and 120 min after injection were measured from the tail-nick blood drops. For ITT, mice were removed from food 2 h before experiment with water provided ad libitum. A solution of regular insulin (Eli Lilly and Company, Indianapolis, IN, USA, 1 U/kg body weight) diluted in normal saline was administrated intraperitoneally. Blood glucose levels at 0, 15, 30, 60, 90, and 120 min after injection were measured. A commercial glucometer (SAFE-ACCU, Shanghai International Holding Corp., Hamburg, Germany) using the glucose oxidase method was used to determined blood glucose levels.

### 4.8. SDS-PAGE and Western Blotting

The cells were washed with 1X PBS and lysed by RIPA buffer (Thermo Fisher Sci-entific Inc., Waltham, MA, USA, Cat. No. 89900) supplemented with protease and phosphatase inhibitors. The in vivo tissue samples were grinded and homogenized by using a glass Dounce homogenizer and then lysed by RIPA buffer supplemented with protease and phosphatase inhibitors. PierceTM Coomassie (Bradford) Protein Assay Kit (Thermo Fisher Scientific Inc., Waltham, MA, USA) was used to determine the total protein concentrations in lysed samples. Standard Western blotting procedure after SDS-PAGE running was performed. The blotting membrane was blocked in 5% non-fat dry milk dissolved in Tris-buffered saline containing 0.1% (*v*/*v*) Tween 20 (TBST) and then incubated with primary antibodies prepared in 3% bovine serum albumin in TBST at 4 °C on a shaker overnight. After washing four times with TBST (5 min each time), the membrane was incubated with anti-rabbit secondary antibody diluted 10,000 times in 3% non-fat dry milk dissolved in TBST. After washing three times with TBST (5 min each time), the blots on the membrane were visualized using the Miracle StarTM Femto Western Blot Detection System (Intron Biotechnology Co., Ltd., Gyeonggi-do, Korea) and X-ray film (UltraCruz^®^ Autoradiography Film, Santa Cruz Biotechnology Inc., Dallas, TX, USA) or the chemiluminescent image analyzer LAS 4000 (General Electric (GE) Healthcare Life Science, Chicago, IL, USA). NIH ImageJ software was used to perform the blots’ densitometry.

### 4.9. RNA Isolation and Real-Time qPCR

Hypothalamic N1 cells were incubated with *p*-Coumaric acid at a final concentration of 10 μM for 24 h. Cells were then harvested and lysed to extract total RNA using Ambion^®^ Life Technologies Trizol reagent (Thermo Fisher Scientific Inc., Waltham, CA, USA). cDNA was synthesized from one microgram (1 μg) of total RNA using High-Capacity cDNA Reverse Transcription Kits (Applied Biosystem, Warrington, UK) in accordance with the manufacturer’s instructions. For real-time PCR, cDNA and primers were prepared with a Power SYBR^®^ Green PCR Master Mix (Applied Biosystem, Warrington, UK) according to the instruction manual. The primer sequences used for real-time Q-PCR were as follows: *Pomc*, forward, 5’-CAGGTCCTGGAGTCCGAC-3’, reverse, 5’-CATGAAGCCACCGTAACG-3’; *Agrp*, forward, 5’-CGGCCACGAACCTCTGTAG-3’, reverse, 5’-CTCATCCCCTGCCTTTGC-3’; *Npy*, forward, 5’-CTACTCCGCTCTGCGACACT-3’, reverse, 5’-AGTGTCTCAGGGCTGGATCTC-3’; and *18S*, forward, 5′-AACCCGTTGAACCCCATT-3′, reverse, 5′-CCATCCAATCGGTAGTAGCG-3′.

### 4.10. Statistical Analysis

Prism 5.0 software was used for all statistical analyses. Two-way ANOVA was used to assess the effects of different time points in body weight tracing, GTT, and ITT analyses. One-way ANOVA was used to analyze data from different treatment groups for blood glucose levels, plasma metabolic hormone levels, and food intake experiments. For other data, Student’s t-test was used. *p* < 0.05 was regarded as a statistically significant difference.

## Figures and Tables

**Figure 1 ijms-22-01431-f001:**
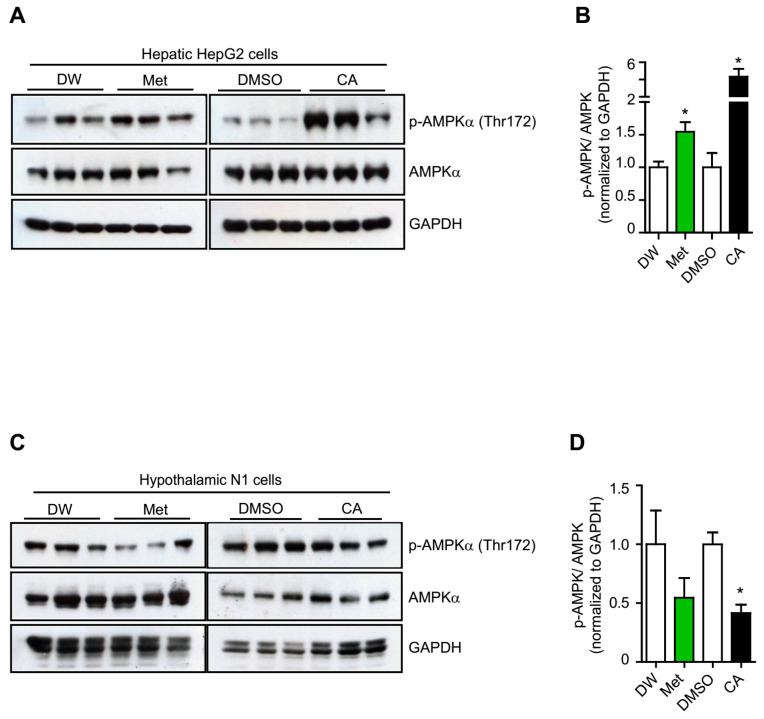
Different effects of CA treatment on AMPK activation in hepatic HepG2 cells and hypothalamic N1 cells. Immunoblots (**A**) and graph (**B**) showing effects of CA and metformin (met) treatments on AMPK activation in the hepatic HepG2 cells. Immunoblots (**C**) and graph (**D**) showing effects of CA and metformin treatments on AMPK activation in the hypothalamic N1 cells. The results are expressed as mean ± SEM. Two-tail Student’s t-test. * *p* < 0.05. DW: distilled water.

**Figure 2 ijms-22-01431-f002:**
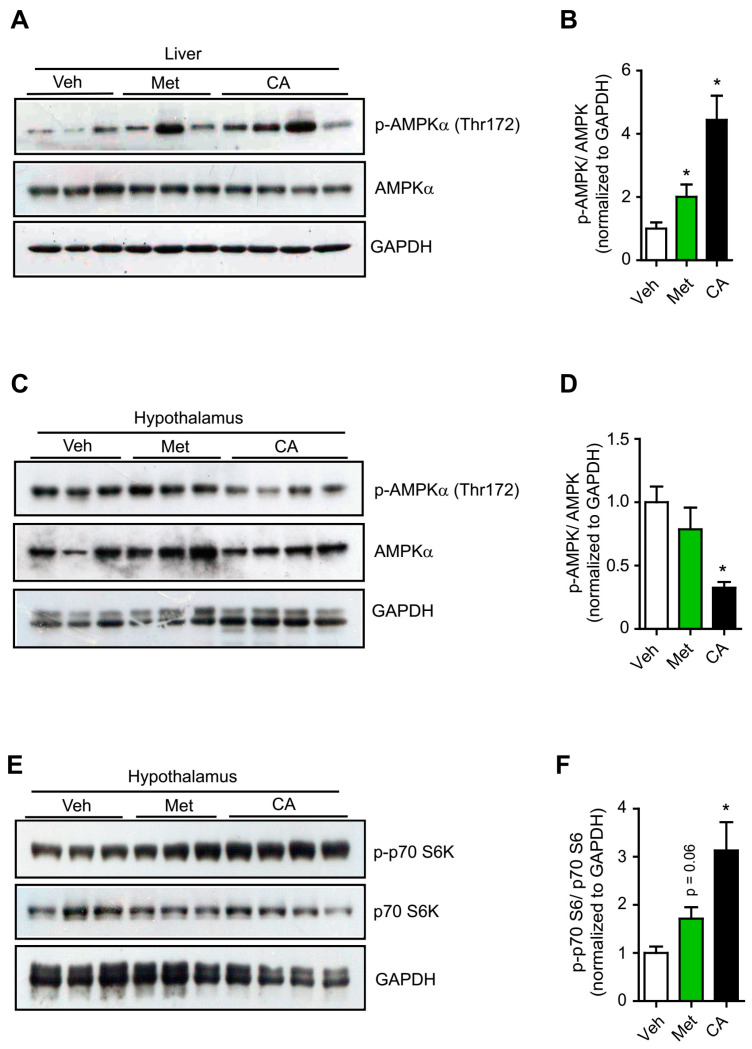
CA differential effects on AMPK activation in vivo. Immunoblots (**A**) and graph (**B**) showing effects of CA and metformin (met) treatments on AMPK activation in the mice livers. Immunoblots (**C**) and graph (**D**) showing effects of CA and metformin treatments on AMPK activation in the hypothalamic samples. Immunoblots (**E**) and graph (**F**) showing effects of CA and metformin treatments on p70S6K activation in the hypothalamic samples. The results are expressed as mean ± SEM. One-way ANOVA with Tukey’s post-hoc tests for comparison of multiple groups or Student’s t-test. * *p* < 0.05. Veh: vehicle.

**Figure 3 ijms-22-01431-f003:**
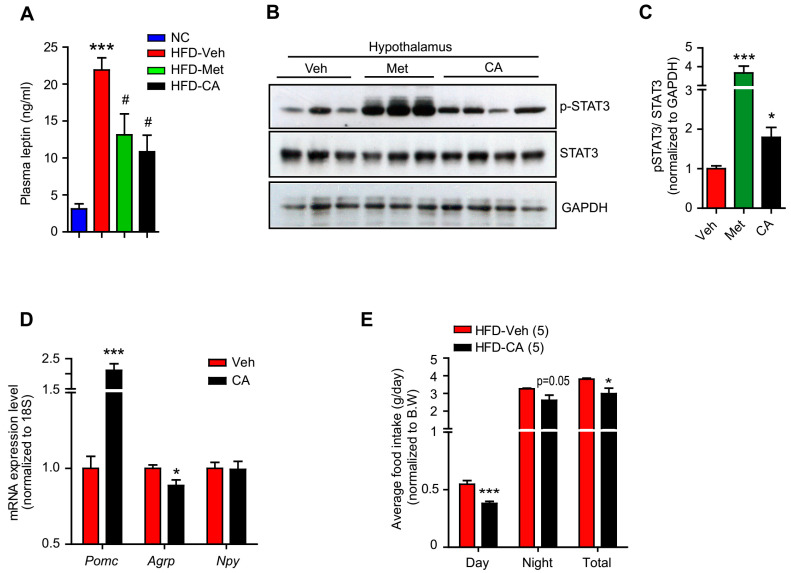
CA treatment enhanced hypothalamic leptin sensitivity. (**A**) Plasma leptin levels of normal chow (NC)-fed mice and high-fat diet (HFD)-fed mice treated with vehicle (veh), CA, and metformin (met). *** *p* < 0.001 compared to NC, # *p* < 0.05 compared to HFD-Veh. Immunoblots (**B**) and graph (**C**) showing effects of vehicle, CA, and metformin treatments on STAT3 downstream signaling in the hypothalamus samples. * *p* < 0.05, *** *p* < 0.001 compared to Veh. (**D**) Transcriptional expression of *Pomc*, *Agrp,* and *Npy* in the hypothalamic N1 cells under vehicle and CA treatments. (**E**) Daily food intake of HFD-fed mice treated with vehicle and CA (averaged from 3 days). The results are expressed as mean ± SEM. One-way ANOVA with Tukey’s post-hoc tests for comparison of multiple groups or Student’s t-test. * *p* < 0.05, *** *p* < 0.001.

**Figure 4 ijms-22-01431-f004:**
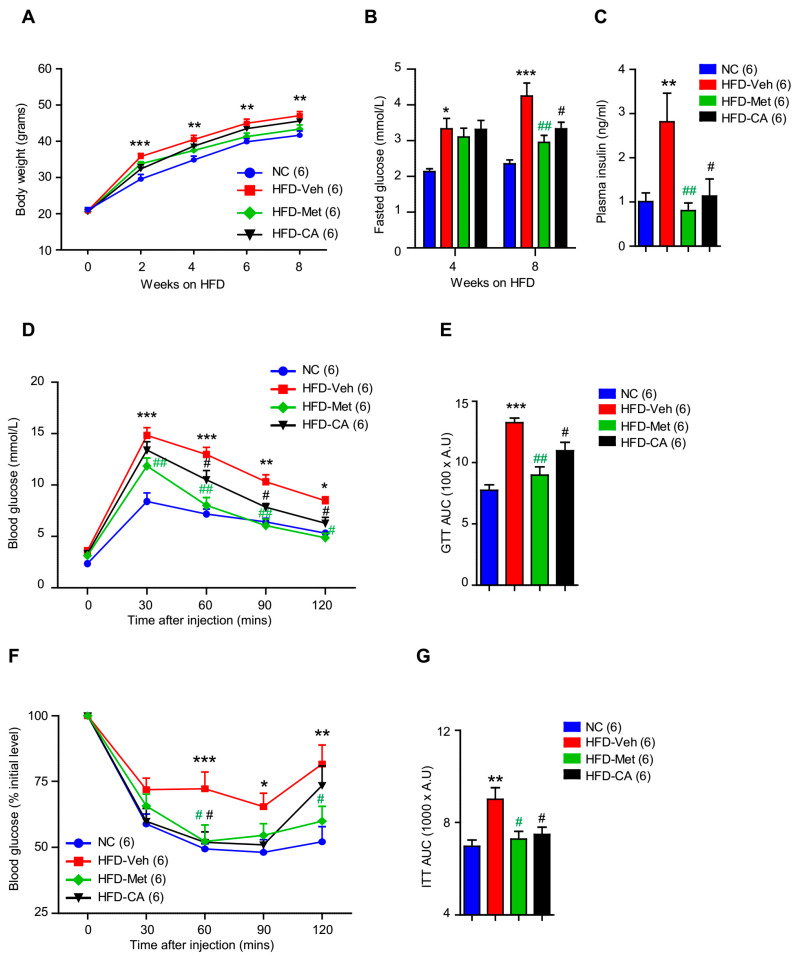
CA treatment improved whole-body glucose homeostasis of mice. (**A**) Body weights of normal chow (NC)-fed mice and high-fat diet (HFD)-fed mice treated with vehicle (veh), metformin (met), and CA for 8 weeks. (**B**) Fasting blood glucose levels of NC-fed mice and HFD-fed mice treated with vehicle, metformin, and CA for 8 weeks. (**C**) Plasma insulin levels of NC-fed mice and HFD-fed mice treated with vehicle, metformin, and CA for 8 weeks. GTT (**D**) and GTT AUC (**E**) of NC-fed mice and HFD-fed mice treated with vehicle, metformin, and CA for 6 weeks. ITT (**F**) and ITT AUC (**G**) of NC-fed mice and HFD-fed mice treated with vehicle, metformin, and CA for 6 weeks. The results are expressed as mean ± SEM. One-way ANOVA with Tukey’s post-hoc tests in bar graphs and two-way ANOVA with Bonferroni post-hoc tests in line graphs. * *p* < 0.05, ** *p* < 0.01, *** *p* < 0.001 compared to NC. # *p* < 0.05, ## *p* < 0.01 compared to HFD-Veh.

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
