# Peer review of "p-Coumaric Acid Enhances Hypothalamic Leptin Signaling and Glucose Homeostasis in Mice via Differential Effects on AMPK Activation"

_ijms, 2021, doi:10.3390/ijms22031431_

Round 1

Reviewer 1 Report

The manuscript “p-coumaric acid enhances hypothalamic leptin signaling and glucose homeostasis in mice via differential effects on AMPK activation” by KW Kim, KV Doan and coworkers provides the molecular and in vivo analysis of effects of hydroxycinnamic phenolic acid on AMPK, p70K and STAT3 phosphorylation, POMC, NPY and AgRP expression in liver and hypothalamic cell lines and mice samples as well as on plasma leptin levels, food intake, body weight, glucose tolerance and insulin sensitivity. The scope of the study is clearly delineated, the methods used are described in sufficient detail, the results are presented in concise yet precise fashion, and the conclusions drawn in the discussion are justified with one exception (see below). Activation of AMPK in the periphery by CA may explain the metabolic effects (improvement of glucose homeostasis) whereas AMPK signaling in the hypothalamus may be responsible for induction of leptin signaling with the resulting significant and non-significant lowering of food intake and body weight, respectively. These findings are of potential interest for readers of the International Journal of Molecular Sciences provided the following points were adequately addressed by the authors.

Major points

  1. A single concentration of CA was used only for the cellular experiments and 20 µM is rather high for a bioactive compound (see 2 µM effective conc. of metformin). A concentration dependence should be presented for the effects on AMPK phosphorylation in HepG2 and N1 cells to gain some insight into the EC50 and IC50 values, respectively.
  2. For treatment of the HFD-mice, a dose of 200 mg/kg was used. It would be very helpful if data about the plasma as well as cerebrospinal fluids were available in order to see whether there is some correlation between the in vitro and in vivo effective concentrations.
  3. Did the authors assay for the effect of CA on p70S6K in cultured HepG2 and N1 cells as well as in liver samples from treated mice? This could provide valuable positive and negative controls.
  4. Discussion, last sentence: ...“our study demonstrated the contribution of central leptin signaling in mediating the metabolic actions of CA on whole-body glucose homeostasis....” The data presented do not support this conclusion. For its demonstration animals, additional experimentation is required, such as the use of animals with defective leptin signaling. Rather, the considerable effects of CA on glucose and insulin tolerance in HFD-mice exhibiting pronounced leptin resistance, even upon administration of CA, argue for a minor role of CA action in the hypothalamus for its upregulation of glucose metabolism.

Minor points

  1. GAPDH was included in the immune blotting analyses, apparently as loading control. Presumably, the different levels of GAPDH have not been considered in the calculation of the p-AMPK/AMPK ratio, since the amounts of p-AMPK and AMPK may be affected in analogous fashion.
  2. Figure 3D: The colors of the columns are mixed up.

Author Response

Response to Reviewer 1 Comments

We sincerely appreciate the Reviewer for taking your valuable time to carefully read and give insightful comments on our manuscript. Here, we would like to address the reviewer’s comments as follows.

Reviewer Comments and Suggestions for Authors

The manuscript “p-coumaric acid enhances hypothalamic leptin signaling and glucose homeostasis in mice via differential effects on AMPK activation” by KW Kim, KV Doan and coworkers provides the molecular and in vivo analysis of effects of hydroxycinnamic phenolic acid on AMPK, p70K and STAT3 phosphorylation, POMC, NPY and AgRP expression in liver and hypothalamic cell lines and mice samples as well as on plasma leptin levels, food intake, body weight, glucose tolerance and insulin sensitivity. The scope of the study is clearly delineated, the methods used are described in sufficient detail, the results are presented in concise yet precise fashion, and the conclusions drawn in the discussion are justified with one exception (see below). Activation of AMPK in the periphery by CA may explain the metabolic effects (improvement of glucose homeostasis) whereas AMPK signaling in the hypothalamus may be responsible for induction of leptin signaling with the resulting significant and non-significant lowering of food intake and body weight, respectively. These findings are of potential interest for readers of the International Journal of Molecular Sciences provided the following points were adequately addressed by the authors.

Major points

  1. A single concentration of CA was used only for the cellular experiments and 20 µM is rather high for a bioactive compound (see 2 µM effective conc. of metformin). A concentration dependence should be presented for the effects on AMPK phosphorylation in HepG2 and N1 cells to gain some insight into the EC50 and IC50 values, respectively.

Response:

We sincerely apologize to the reviewer for a formatting mistake in the metformin concentration unit in our original manuscript, which led to the reviewer’s confusion about the metformin concentration used in our study. We used metformin at a final concentration of 2 mM (millimolar), but not 2 mM (micromolar). This dose of metformin was based on our previous study (Doan et al., 2015) and others (Howell et al., 2017; Nelson et al., 2012). For the treatment dose of p-coumaric (CA), we had done dose-dependent experiments to investigate effects of CA on AMPK phosphorylation in HepG2 and N1 cells. Based on the results from dose-dependent experiments, we chose 10 mM or 20 mM of final concentrations of CA treatment for further experiments in N1 or HepG2 cells, respectively. We have provided the results of these dose-dependent experiments in the Supplemental Figure 1A and B and corrected the concentration units of metformin and CA in the Materials and Methods section of the revised manuscript. We have also updated the Results section to incorporate these results accordingly (lines 75-86).

We appreciate the reviewer for this comment to improve our manuscript.

  1. For treatment of the HFD-mice, a dose of 200 mg/kg was used. It would be very helpful if data about the plasma as well as cerebrospinal fluids were available in order to see whether there is some correlation between the in vitro and in vivo effective concentrations.

Response:

We support the reviewer’s view that the determination of CA levels in blood and cerebrospinal fluids (CSF) will give direct evidence for the correlation between the in vitro and in vivo effective concentrations in our study. Unfortunately, we don’t have these data currently. Although we could not provide a direct measurement of CA concentrations in the blood and CSF for the current time, we have summarized the pharmacokinetic data of CA in rodents in previous studies to illustrate that an oral dose of 200 mg/kg of CA would produce CA plasma concentrations at steady-state comparable with in vitro effective concentrations used in our study as follows.

Pharmacokinetic studies in rodents showed that CA was absorbed rapidly in the intestine to reach maximum plasma concentration (Cmax) in a few minutes (4-10 minutes) after oral administration ((Konishi et al., 2004; Meng et al., 2006); reviewed in (Pei et al., 2016)). In mouse, Meng et al. calculated that after the oral administration of a plant extract at a dose containing 11.51 mg/kg of CA, Cmax in the plasma was 35 ± 11 µmol/L and rapidly declined afterward (Meng et al., 2006). Gasperotti et al. comprehensively investigated the pharmacokinetics of 23 polyphenol metabolites including CA following an intravenous dose in rats and found that I.V injection of 15 nmol of CA (as rats had an estimated blood volume of 15 mL (Lee and Blaufox, 1985), the initial blood concentration would be approximately 1 µmol/L) resulted in a steady-state plasma concentration of about 0.04 µmol/L (Gasperotti et al., 2015). Based on these pharmacokinetic data and assume that Cmax is proportional to the total dose, an oral dose of 200 mg/kg of CA would produce a Cmax of around 600 ± 190 µmol/L and result in a steady-state plasma concentration ranging from 15 to 35 mM. Moreover, although detailed pharmacokinetic data of CA levels in the CSF is unavailable, CA’s ability to cross the blood-brain barrier and enter the brain was demonstrated in both in silico and in vivo studies ((Gasperotti et al., 2015); reviewed in (Carregosa et al., 2020)).

We have included these pharmacokinetic studies as references in support for a correlation between the in vitro and in vivo effective concentrations following a 200 mg/kg oral dose of CA in our study. We have also updated the Result and Discussion sections accordingly in the revised manuscript (lines 97-99, 218-224). We appreciate the reviewer for this insightful comment.

  1. Did the authors assay for the effect of CA on p70S6K in cultured HepG2 and N1 cells as well as in liver samples from treated mice? This could provide valuable positive and negative controls.

Response:

We have checked levels of p-p70S6K and p70S6K in samples of cultured HepG2 and N1 cells treated with metformin or CA as well as in liver samples from treated mice. We found that p-p70S6K/p70S6K ratio was increased under CA treatment in N1 cells, similar to metformin’s effect. However, unlike the pronounced inhibition of p-p70S6K/p70S6K ratio under metformin treatment in HepG2 cells and liver samples, CA treatment insignificantly altered the p-p70S6K/p70S6K ratio in HepG2 cells in mice livers. We have provided these results in Supplemental Figure 2A-F and updated the Result section accordingly in the revised manuscript (lines 113-132). We have also discussed about these results in the Discussion section (lines 213-215). We thank the reviewer for this insightful and constructive suggestion.

  1. Discussion, last sentence: ...“our study demonstrated the contribution of central leptin signaling in mediating the metabolic actions of CA on whole-body glucose homeostasis....” The data presented do not support this conclusion. For its demonstration animals, additional experimentation is required, such as the use of animals with defective leptin signaling. Rather, the considerable effects of CA on glucose and insulin tolerance in HFD-mice exhibiting pronounced leptin resistance, even upon administration of CA, argue for a minor role of CA action in the hypothalamus for its upregulation of glucose metabolism.

Response:

We agree with the reviewer that further experiments using mice defected in leptin signaling (db/db mice) are required to demonstrate the contribution of central leptin signaling in mediation of CA’s metabolic actions on whole-body glucose metabolism. Unfortunately, we don’t have any study group to pursue these experiments at the current time. We have thus discussed more on the role of CA action in the central leptin signaling in contribution to its beneficial effects on glucose metabolism as follows.

There is a reciprocal relationship between peripheral AMPK activation and central leptin signaling which are both known to play crucial roles in the regulation of whole-body energy homeostasis (da Silva et al., 2020; Garcia and Shaw, 2017). While peripheral AMPK activation by calorie restriction, exercise, or metformin helps to ameliorate central leptin resistance (Kim et al., 2006; Santoro et al., 2015), central leptin action has been shown to increase the peripheral AMPK (Roman et al., 2010). Moreover, central leptin was shown to inhibit AMPK activity in the hypothalamus to suppress food intake and regulate body weight (Minokoshi et al., 2004). In this regard, beneficial effects of CA on whole-body glucose homeostasis observed in our study could be attributed to its differential effects on peripheral and central AMPK activation or central leptin signaling enhancement or combined actions. Further study using mice with defective leptin signaling may help to demonstrate the contribution of central leptin signaling in mediation of CA’s metabolic actions on whole-body glucose metabolism.

We have modified that last sentence in the Discussion section to properly clarify our findings and included the above discussion in the revised manuscript (lines 250-261, 270-271). We sincerely appreciate the reviewer for this very insightful and constructive comment.

Minor points

  1. GAPDH was included in the immune blotting analyses, apparently as loading control. Presumably, the different levels of GAPDH have not been considered in the calculation of the p-AMPK/AMPK ratio, since the amounts of p-AMPK and AMPK may be affected in analogous fashion.

Response:

We have re-performed the densitometry using p-AMPK/AMPK ratios normalized to GAPDH levels. The results were consistent with the original conclusion that CA treatment increased p-AMPK/AMPK ratio in HepG2 cells and liver samples while decreased AMPK/AMPK ratio in N1 cells and hypothalamic samples. We have updated bar graphs of the densitometry using p-AMPK/AMPK ratios normalized to GAPDH level and modified Figure 1B and D, Figure 2B and D, and Figure 3C accordingly in the revised manuscript.

  1. Figure 3D: The colors of the columns are mixed up.

Response:

We sincerely apologize to the reviewer for our mistake on the colors of bar graphs in Figure 3D. The black bar should be Veh treatment and the red bar should be CA treatment. We have corrected the colors of bar graph columns in the Figure 3D in the revised manuscript.

We appreciate the reviewer for a careful reading and insightful comments on our manuscript.

Sincerely,

Khanh V. Doan, Ph.D.,

School of Pharmacy, Van Lang University.

Address: 233A Phan Van Tri, Ward 11, Binh Thanh District, Ho Chi Minh City, 700000, Viet Nam.

Tel: (+84-28)-7109-9284 Ext 4210

Cell: (+84-91)-185-1177

References

Carregosa, D., Carecho, R., Figueira, I., and C, N.S. (2020). Low-Molecular Weight Metabolites from Polyphenols as Effectors for Attenuating Neuroinflammation. Journal of agricultural and food chemistry 68, 1790-1807.

da Silva, A.A., do Carmo, J.M., and Hall, J.E. (2020). CNS Regulation of Glucose Homeostasis: Role of the Leptin-Melanocortin System. Current diabetes reports 20, 29.

Doan, K.V., Ko, C.M., Kinyua, A.W., Yang, D.J., Choi, Y.H., Oh, I.Y., Nguyen, N.M., Ko, A., Choi, J.W., Jeong, Y., et al. (2015). Gallic acid regulates body weight and glucose homeostasis through AMPK activation. Endocrinology 156, 157-168.

Garcia, D., and Shaw, R.J. (2017). AMPK: Mechanisms of Cellular Energy Sensing and Restoration of Metabolic Balance. Molecular cell 66, 789-800.

Gasperotti, M., Passamonti, S., Tramer, F., Masuero, D., Guella, G., Mattivi, F., and Vrhovsek, U. (2015). Fate of microbial metabolites of dietary polyphenols in rats: is the brain their target destination? ACS chemical neuroscience 6, 1341-1352.

Howell, J.J., Hellberg, K., Turner, M., Talbott, G., Kolar, M.J., Ross, D.S., Hoxhaj, G., Saghatelian, A., Shaw, R.J., and Manning, B.D. (2017). Metformin Inhibits Hepatic mTORC1 Signaling via Dose-Dependent Mechanisms Involving AMPK and the TSC Complex. Cell metabolism 25, 463-471.

Kim, Y.W., Kim, J.Y., Park, Y.H., Park, S.Y., Won, K.C., Choi, K.H., Huh, J.Y., and Moon, K.H. (2006). Metformin restores leptin sensitivity in high-fat-fed obese rats with leptin resistance. Diabetes 55, 716-724.

Konishi, Y., Hitomi, Y., and Yoshioka, E. (2004). Intestinal absorption of p-coumaric and gallic acids in rats after oral administration. Journal of agricultural and food chemistry 52, 2527-2532.

Lee, H.B., and Blaufox, M.D. (1985). Blood volume in the rat. Journal of nuclear medicine : official publication, Society of Nuclear Medicine 26, 72-76.

Meng, Z., Wang, W., Xing, D.M., Lei, F., Lan, J.Q., and Du, L.J. (2006). Pharmacokinetic study of p-coumaric acid in mouse after oral administration of extract of Ananas comosus L. leaves. Biomedical chromatography : BMC 20, 951-955.

Minokoshi, Y., Alquier, T., Furukawa, N., Kim, Y.B., Lee, A., Xue, B., Mu, J., Foufelle, F., Ferre, P., Birnbaum, M.J., et al. (2004). AMP-kinase regulates food intake by responding to hormonal and nutrient signals in the hypothalamus. Nature 428, 569-574.

Nelson, L.E., Valentine, R.J., Cacicedo, J.M., Gauthier, M.S., Ido, Y., and Ruderman, N.B. (2012). A novel inverse relationship between metformin-triggered AMPK-SIRT1 signaling and p53 protein abundance in high glucose-exposed HepG2 cells. American journal of physiology Cell physiology 303, C4-C13.

Pei, K., Ou, J., Huang, J., and Ou, S. (2016). p-Coumaric acid and its conjugates: dietary sources, pharmacokinetic properties and biological activities. Journal of the science of food and agriculture 96, 2952-2962.

Roman, E.A., Reis, D., Romanatto, T., Maimoni, D., Ferreira, E.A., Santos, G.A., Torsoni, A.S., Velloso, L.A., and Torsoni, M.A. (2010). Central leptin action improves skeletal muscle AKT, AMPK, and PGC1 alpha activation by hypothalamic PI3K-dependent mechanism. Molecular and cellular endocrinology 314, 62-69.

Santoro, A., Mattace Raso, G., and Meli, R. (2015). Drug targeting of leptin resistance. Life sciences 140, 64-74.

Reviewer 2 Report

Advantages good writing and scientific structures of the context. There are some problems, which is needed to be solved 1. The authors did not clearly explain why you choose HepG2 cells as one of your targets in the context. Another concern is that HepG2 is a liver cancer cells, and the N1 is an embryonic neurons. Do you think that your findings in totally different cell lines can apply to an in vivo study? Although your results are somewhat convincible, however, it is a little bit far-fetched for your readers to follow your thinking and most of them may ask "why you chose those cells, and amazingly, your findings can be perfectly explained an in vivo setting"? 2. Please explain figure 2 c, d. How the authors can select "only neurons" from a hypothalamus? because the region is apparently heterogenous. Or your response represent all kind of cells in the hypothalamus? but the cell study you showed in the figure 1 is neuron? 3. Figure 3A is interesting, and I suggest a CSF data may be better. 4. Figure 3 B: It is apparently that effects of Met on stat3 are stronger than CA's. Do you think that the mechanisms of CA on leptin may be different from met on leptin? 5. Line 125-128 and line 201-204 "Furthermore, CA treatment enhanced expression of the anorexigenic neuropeptide proopiomelanocortin (Pomc) while inhibited expression of the orexigenic agouti-related neuropeptide (Agrp) which are well-known target genes of the hypothalamic leptin- STAT3 signaling (Figure 3D)" Your statement is not compatible with the figure 3D. My understanding is that the Veh enhanced expression of Pomc. The statistical significance in the agrp needs to be rechecked.

Author Response

Response to Reviewer 2 Comments

We sincerely appreciate the Reviewer for taking your valuable time to carefully read and give insightful comments on our manuscript. Here, we would like to address the reviewer’s comments as follows.

Reviewer Comments and Suggestions for Authors

Advantages good writing and scientific structures of the context. There are some problems, which is needed to be solved:

  1. The authors did not clearly explain why you choose HepG2 cells as one of your targets in the context. Another concern is that HepG2 is a liver cancer cells, and the N1 is an embryonic neurons. Do you think that your findings in totally different cell lines can apply to an in vivo study? Although your results are somewhat convincible, however, it is a little bit far-fetched for your readers to follow your thinking and most of them may ask "why you chose those cells, and amazingly, your findings can be perfectly explained an in vivo setting"?

Response:

We have updated the Results section in the revised manuscript to explain the reason for choosing hepatic HepG2 and hypothalamic neuronal N1 cells to investigate CA’s effect on AMPK activation in our study (lines 73-80). In summary, the reference drug metformin is well-known to activate AMPK prominently in hepatocytes to exert its therapeutic actions (Shaw et al., 2005; Zhou et al., 2001). However, recent studies indicated that metformin also acted on the hypothalamic neurons in the central nervous system (CNS) to reduce food intake and these anorexigenic actions of metformin were linked to its inhibitory effect on AMPK activity (Chau-Van et al., 2007; Stevanovic et al., 2012).

We agree that primary culture of hepatocytes and hypothalamic neurons would be the best in vitro models to mimic in vivo settings, however, immortalized cell lines (HepG2, N1) which are commonly used in many laboratories was also acceptable to investigate the molecular and intracellular mechanisms in metabolic studies (Dalvi et al., 2011; Sefried et al., 2018). Moreover, as the reviewer noted, the in vitro data in our study were solid and convincing as we used metformin as the positive control in every experiment and more importantly, the in vitro data were supported by consistent in vivo findings.

We appreciate the reviewer for this constructive comment which helped us to clarify the cell lines used in our study.

  1. Please explain figure 2 c, d. How the authors can select "only neurons" from a hypothalamus? because the region is apparently heterogenous. Or your response represent all kind of cells in the hypothalamus? but the cell study you showed in the figure 1 is neuron?

Response:

We collected intact hypothalamus which contained both neurons and non-neuronal cells from treated mice and measured levels of p-AMPK (Thr172) and total AMPK in these homogenized hypothalamic samples by Western blot (Figure 2C). The densitometry results showed that the ratio of p-AMPK/AMPK in hypothalamic samples of CA-treated mice was significantly decreased compared to that of Veh-treated mice (Figure 2D). This result indicated that CA treatment inhibited hypothalamic AMPK activation in vivo.

We agree with the reviewer that embryonic hypothalamic neuronal cells used in the Figure 1C and D could not represent all types of cells in the hypothalamus. However, hypothalamic cell lines have been effectively used to investigate the AMPK-dependent physiological processes underlying central glucose sensing in the hypothalamus (Cai et al., 2007), indicating that hypothalamic cell lines are an appropriate in vitro model for studying the molecular and intracellular mechanisms of hypothalamic function (Dalvi et al., 2011). These immortalized hypothalamic cell lines and in vivo models have also been suggested to be complementary to each other to provide great potential means in drug discovery and therapeutic drug testing (Dalvi et al., 2011). We therefore postulated that the inhibitory effect of CA treatment on AMPK activity observed in the hypothalamic samples from treated mice recapitulated that of CA treatment in the cultured hypothalamic N1 neurons.

  1. Figure 3A is interesting, and I suggest a CSF data may be better.

Response:

We support the reviewer’s view that leptin levels in the CSF will provide better evidence for the central leptin action. Unfortunately, we don’t have data on leptin levels in the CSF samples currently. However, CSF leptin has been shown to strongly correlate with plasma leptin levels (Schwartz et al., 1996). In addition, reduced plasma leptin level under CA treatment was accompanied with increased p-STAT3 level and leptin/STAT3 signaling effect on the target genes (Pomc and Agrp) in the hypothalamus as well as on the food intake. These results highly indicated an enhancement of hypothalamic leptin signaling under CA treatment.

  1. Figure 3 B: It is apparently that effects of Met on stat3 are stronger than CA's. Do you think that the mechanisms of CA on leptin may be different from met on leptin?

Response:

This is an excellent question and we agree with the reviewer that based on the result from Figure 3B, the effect of metformin on STAT3, a leptin downstream effector, was more pronounced than that of CA treatment. The effect of metformin and CA on STAT3 might be indirect and resulted from enhanced leptin signaling as we did not observe any change in p-STAT3 levels in N1 cells under metformin or CA treatment (data not shown). Based on effects of metformin and CA treatments on mice body weight (Figure 4A), we assumed that leptin signaling might be affected by metformin treatment at a greater magnitude than by CA treatment. However, further studies are needed to distinguish the mechanisms of CA and metformin actions on leptin signaling, which may be beyond of the scope of our current study. We thank the reviewer for this insightful and interesting suggestion.

  1. Line 125-128 and line 201-204 "Furthermore, CA treatment enhanced expression of the anorexigenic neuropeptide proopiomelanocortin (Pomc) while inhibited expression of the orexigenic agouti-related neuropeptide (Agrp) which are well-known target genes of the hypothalamic leptin- STAT3 signaling (Figure 3D)" Your statement is not compatible with the figure 3D. My understanding is that the Veh enhanced expression of Pomc. The statistical significance in the agrp needs to be rechecked.

Response:

We sincerely apologize to the reviewer for our mistake on the colors of bar graphs in the Figure 3D. The black bar should be Veh treatment and the red bar should be CA treatment. We have corrected the colors of bar graph columns in Figure 3D of the revised manuscript.

For the statistical significance in the Agrp relative gene expression, we have re-checked the raw data and confirmed that the difference between two groups was minimal, but significant. (Veh: 1.000 ± 0.02274, N=6; and CA: 0.8873 ± 0.03653, N=6; P =0.0256, Two-tailed Student’s t-test). We have transformed the Y-axis with two-segments in Figure 3D for better illustration of the results and updated Figure 3D correspondingly.

We thank the reviewer for this suggestion.

We appreciate the reviewer for a careful reading and insightful comments on our manuscript.

Sincerely,

Khanh V. Doan, Ph.D.,

School of Pharmacy, Van Lang University.

Address: 233A Phan Van Tri, Ward 11, Binh Thanh District, Ho Chi Minh City, 700000, Viet Nam.

Tel: (+84-28)-7109-9284 Ext 4210

Cell: (+84-91)-185-1177

References

Cai, F., Gyulkhandanyan, A.V., Wheeler, M.B., and Belsham, D.D. (2007). Glucose regulates AMP-activated protein kinase activity and gene expression in clonal, hypothalamic neurons expressing proopiomelanocortin: additive effects of leptin or insulin. The Journal of endocrinology 192, 605-614.

Chau-Van, C., Gamba, M., Salvi, R., Gaillard, R.C., and Pralong, F.P. (2007). Metformin inhibits adenosine 5'-monophosphate-activated kinase activation and prevents increases in neuropeptide Y expression in cultured hypothalamic neurons. Endocrinology 148, 507-511.

Dalvi, P.S., Nazarians-Armavil, A., Tung, S., and Belsham, D.D. (2011). Immortalized neurons for the study of hypothalamic function. American journal of physiology Regulatory, integrative and comparative physiology 300, R1030-1052.

Schwartz, M.W., Peskind, E., Raskind, M., Boyko, E.J., and Porte, D., Jr. (1996). Cerebrospinal fluid leptin levels: relationship to plasma levels and to adiposity in humans. Nature medicine 2, 589-593.

Sefried, S., Haring, H.U., Weigert, C., and Eckstein, S.S. (2018). Suitability of hepatocyte cell lines HepG2, AML12 and THLE-2 for investigation of insulin signalling and hepatokine gene expression. Open biology 8.

Shaw, R.J., Lamia, K.A., Vasquez, D., Koo, S.H., Bardeesy, N., Depinho, R.A., Montminy, M., and Cantley, L.C. (2005). The kinase LKB1 mediates glucose homeostasis in liver and therapeutic effects of metformin. Science 310, 1642-1646.

Stevanovic, D., Janjetovic, K., Misirkic, M., Vucicevic, L., Sumarac-Dumanovic, M., Micic, D., Starcevic, V., and Trajkovic, V. (2012). Intracerebroventricular administration of metformin inhibits ghrelin-induced Hypothalamic AMP-kinase signalling and food intake. Neuroendocrinology 96, 24-31.

Zhou, G., Myers, R., Li, Y., Chen, Y., Shen, X., Fenyk-Melody, J., Wu, M., Ventre, J., Doebber, T., Fujii, N., et al. (2001). Role of AMP-activated protein kinase in mechanism of metformin action. The Journal of clinical investigation 108, 1167-1174.

Round 2

Reviewer 1 Report

The authors have met my criticism in adequate an consequent fashion.

Thank you!